# Monitoring transmission intensity of trachoma with serology

Christine Tedijanto[1], Anthony W. Solomon[2], Diana L. Martin[3], Scott D. Nash[4], Jeremy D. Keenan[1,5], Thomas M. Lietman[1,5,6,7], Patrick J. Lammie[8], Kristen Aiemjoy[9], Abdou Amza[10,11], Solomon Aragie[1,12,13], Ahmed M. Arzika[14], E. Kelly Callahan[4], Sydney Carolan[1], Adisu Abebe Dawed[15], E. Brook Goodhew[3], Sarah Gwyn[3], Jaouad Hammou[16], Boubacar Kadri[10,11], Khumbo Kalua[17], Ramatou Maliki[14], Beido Nassirou[10,11], Fikre Seife[18], Zerihun Tadesse[12], Sheila K. West[19], Dionna M. Wittberg[1], Taye Zeru Tadege[20] & Benjamin F. Arnold[1,5] ✉

Trachoma, caused by ocular *Chlamydia trachomatis* infection, is targeted for global elimination as a public health problem by 2030. To provide evidence for use of antibodies to monitor *C. trachomatis* transmission, we collated IgG responses to Pgp3 antigen, PCR positivity, and clinical observations from 19,811 children aged 1–9 years in 14 populations. We demonstrate that age-seroprevalence curves consistently shift along a gradient of transmission intensity: rising steeply in populations with high levels of infection and active trachoma and becoming flat in populations near elimination. Seroprevalence (range: 0–54%) and seroconversion rates (range: 0–15 per 100 person-years) correlate with PCR prevalence (r: 0.87, 95% CI: 0.57, 0.97). A seroprevalence threshold of 13.5% (seroconversion rate 2.75 per 100 person-years) identifies clusters with any PCR-identified infection at high sensitivity (>90%) and moderate specificity (69–75%). Antibody responses in young children provide a robust, generalizable approach to monitor population progress toward and beyond trachoma elimination.

Trachoma is the leading infectious cause of blindness and has been targeted for elimination as a public health problem by 2030[1]. Caused by repeated infections with the bacterium *Chlamydia trachomatis*, trachoma has historically been monitored via district-level estimates of clinical signs[2]. However, clinical signs may not align well with transmission intensity, particularly after mass drug administration of azithromycin (MDA)[3–5], and are prone to measurement error, especially as cases become rare[6–8]. As a result, there has been growing interest in IgG antibody responses as an objective and easy-to-collect biomarker to inform control programs. Based on examples from other infectious diseases including malaria[9] and SARS-CoV-2[10], use cases for trachoma serology may include monitoring population-level transmission to determine appropriate interventions, post-validation surveillance,

opportunistic investigation of populations using blood collected for other purposes, and fine-scale geographic targeting of control measures[11–13].

IgG antibody responses to *C. trachomatis* antigens Pgp3 and CT694 are sensitive and specific markers of *C. trachomatis* infection[14,15]. Assays have demonstrated consistency across platforms (multiplex bead, enzyme-linked immunosorbent, lateral flow)[16–18], robust seropositivity cutoffs[19], and high repeatability[18,20]. Of the two antigens, Pgp3 appears to elicit a stronger and more durable response and is collected in most trachoma serology surveys[21]. In epidemiologic studies, investigators have observed alignment between population-level summaries of antibody response and other trachoma markers, including the presence of *C. trachomatis* DNA, as determined by PCR,

and the active trachoma sign trachomatous inflammation−follicular (TF). Individual studies suggest age-dependent seroprevalence curves rise steeply with age among children in trachoma-endemic communities[14,15] and are flat in communities that have eliminated trachoma[22–24]. IgG responses to Pgp3 can be measured from dried blood spots and have the potential to be scaled through integration in multiplex surveillance strategies with other infectious pathogens[25]. Key next steps to advance the use of serology to monitor trachoma programs are to benchmark serology against independent measures of transmission, notably PCR, and to formalize the analytic approaches used to summarize trachoma antibody data.

Here, we collated IgG antibody, infection, and clinical data from 14 populations in five countries across a gradient of trachoma transmission intensities that ranged from hyper-endemic to post-elimination. Due to the shortcomings of clinical signs, we used PCR as the primary marker of trachoma activity, with TF as a secondary indicator to

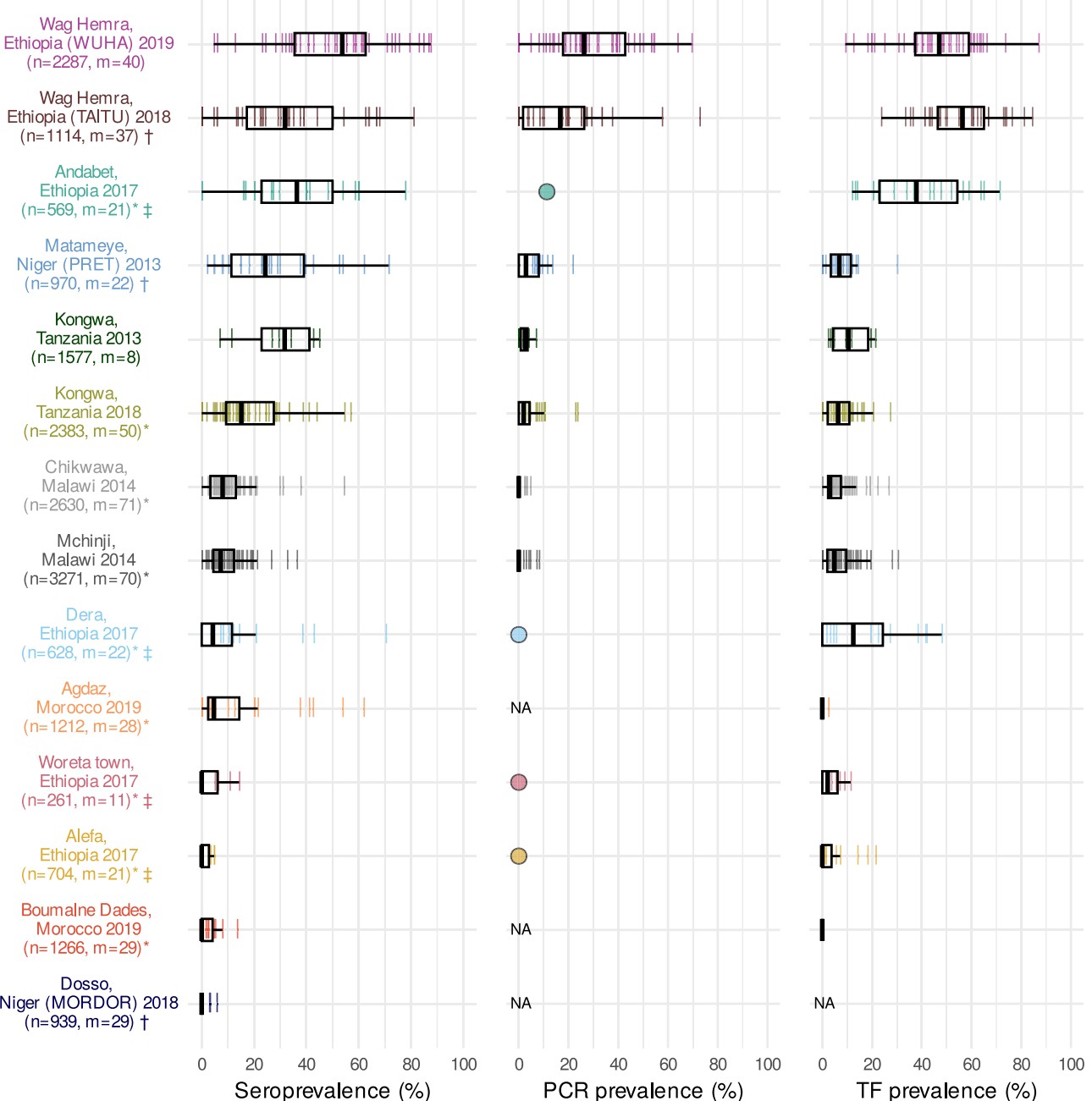

**Fig. 1 | Seroprevalence, PCR positivity, and TF prevalence across study populations.** Cluster-level prevalence estimates are represented by colored lines, and overlaid boxplots show the median, interquartile range, and range (excluding outliers defined as points more than 1.5 times the interquartile range from the 25th or 75th percentile) for each population. Study site, country, study name (if applicable), and year of data collection are listed on the left with number of individuals (*n*) and number of clusters (*m*) included in the analyses. For studies with only district-level estimates of PCR prevalence the mean is indicated with a circle rather than box plot. NA marks studies that did not measure PCR or trachomatous inflammation−follicular (TF). Study populations are arranged in descending order of seroconversion rates, presented in Fig. 2. Source data are provided with this paper. Created with notebook https://osf.io/qjst8.

\* Estimates collected in population based survey
† Serology, PCR, and TF measured among 1-5 year olds only
‡ PCR prevalence measured only at the district-level among 1-5 year olds

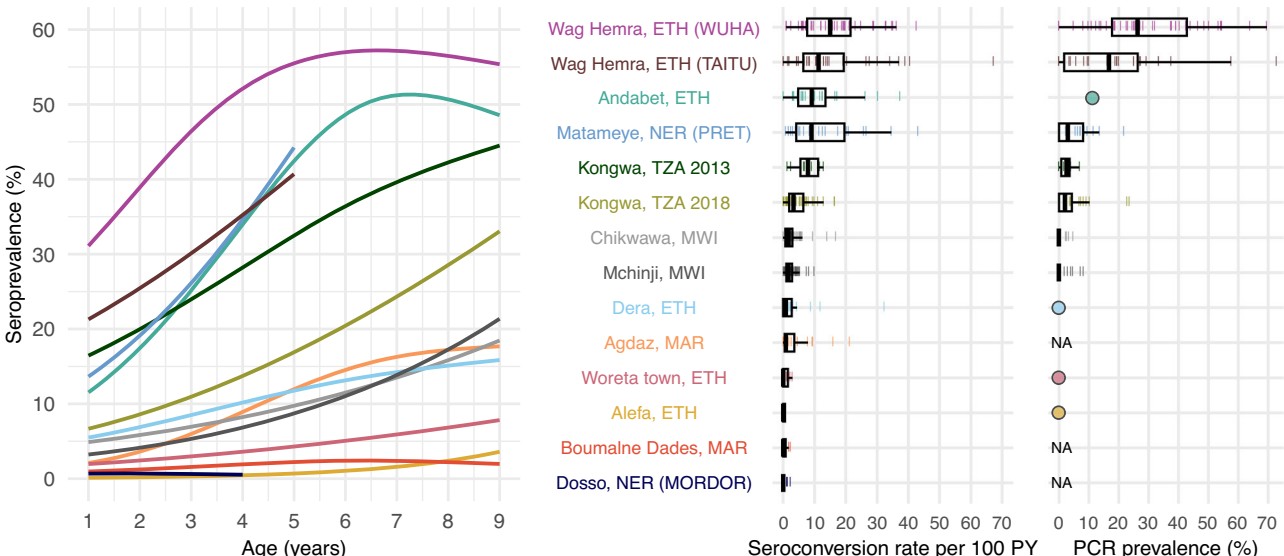

**Fig. 2 | Age-dependent seroprevalence curves, modeled seroconversion rates, and PCR prevalence across study populations.** Cubic splines fit to seroprevalence by age are shown in the left plot. Cluster-level seroconversion rates and PCR prevalence estimates are represented by colored lines, and overlaid boxplots show the median, interquartile range, and range (excluding outliers defined as points >1.5 times the interquartile range from the 25th or 75th percentile) for each population. Figure 1 includes each study's sample size. Study populations are arranged in descending order of median seroconversion rate assuming no seroreversion. *C. trachomatis* PCR measurements are identical to Fig. 1 and are included for reference. For studies with only district-level estimates of PCR prevalence, the mean is indicated with a circle rather than box plot. NA indicates that a study did not measure *C. trachomatis* infections with PCR. *PY* person-years, *ETH* Ethiopia, *MAR* Morocco, *MWI* Malawi, *NER* Niger, *TZA* Tanzania. Source data are provided with this paper. Created with notebook https://osf.io/rghyc.

contextualize serologic estimates. We hypothesized that population-level summaries of trachoma antibody measurements would follow a consistent path toward elimination, characterized by flattening of age-seroprevalence curves and a corresponding decline in seroconversion rates. We also hypothesized that different seroepidemiologic summary measures (e.g., seroprevalence, seroconversion rates) would be consistent with one another and align similarly with population-level transmission. We completed a series of analyses to test these hypotheses and in doing so, develop robust analytic methods for trachoma serology as we approach the elimination endgame.

## Results

### Study populations and settings

We combined serologic, molecular, and clinical measurements collected between 2012 and 2019 across nine studies in five countries (Supplementary Table 1)[26–38]. When possible, measurements within the same study were stratified into populations corresponding to trachoma evaluation units, resulting in 14 study populations. All studies were conducted in Africa and represented a gradient of trachoma transmission from hyper-endemic transmission to post-elimination. Measurements were taken at a single time point or in repeated cross-sectional visits (typically annual measurements). Due to changing transmission and/or interventions over time in some studies, analyses included only the most recent round of measurements for each population. We analyzed sampling clusters with 15 or more serologic measurements, a total of 459 clusters. The median number of children contributing information per cluster was 40 (interquartile range: 32–47) (Supplementary Fig. 1), for a total of 19,811 serologic measurements. Eleven studies included children aged 1–9 years, and three studies focused on children aged 1–5 years (Supplementary Fig. 2).

We generated population-level summaries using the median and range across clusters and observed wide variation in seroprevalence (median: 8%; range: 0–54%), PCR prevalence (median: 3%; range: 0–26%), and TF prevalence (median: 6%; range: 0–56%) (Fig. 1). PCR, a measure of current infection, may disappear rapidly following MDA[39]. Among clusters with MDA in the past year, the median PCR prevalence

was 0%, while among clusters without recent MDA, PCR prevalence was 12%. Cluster-level estimates within the same population were often variable, likely a combination of heterogeneity in disease transmission and stochasticity due to smaller sample sizes. Across the gradient of transmission intensities, population-level summaries of seroprevalence aligned with PCR and TF prevalence (Fig. 1).

### Charting progress toward elimination with serology

We estimated age-dependent seroprevalence curves using semiparametric splines and seroconversion rates from age-structured seroprevalence (Methods). Across populations, seroconversion rates ranged from 0 to 15 per 100 person-years (median: 1.7) using a catalytic model without seroreversion (Fig. 2). Consistent with our hypothesis, we observed that age-seroprevalence curves were flatter and seroconversion rates were lower in areas with lower PCR prevalence. Age-seroprevalence curves rose steeply in settings with high levels of infection such as Wag Hemra and Andabet, Ethiopia, reaching >50% seroprevalence by age 9 years. In contrast, in populations with flatter curves (<10% seroprevalence by age 9 years, seroconversion rate <1 per 100 person-years), PCR prevalence was 0% when measured (Woreta town and Alefa, Ethiopia). Among populations with moderately increasing age-seroprevalence curves, roughly corresponding to seroconversion rates between 1 and 5 per 100 person-years, PCR prevalence ranged from 0 (Dera, Ethiopia) to 1.9% (Kongwa, Tanzania 2018).

### Agreement between serology-based summary measures

Summary statistics estimated from age-seroprevalence curves provide additional, useful summary measures of transmission[9,40]. Population-level summaries of IgG responses include geometric mean IgG levels, seroprevalence, and seroconversion rates—each progression from IgG levels to seroconversion rates relies on additional model assumptions and analysis complexity (details in Methods). In the estimation of seroconversion rates from age-structured seroprevalence, we considered a range of model complexity that included a catalytic model (corresponding to a susceptible-infected-recovered, SIR,

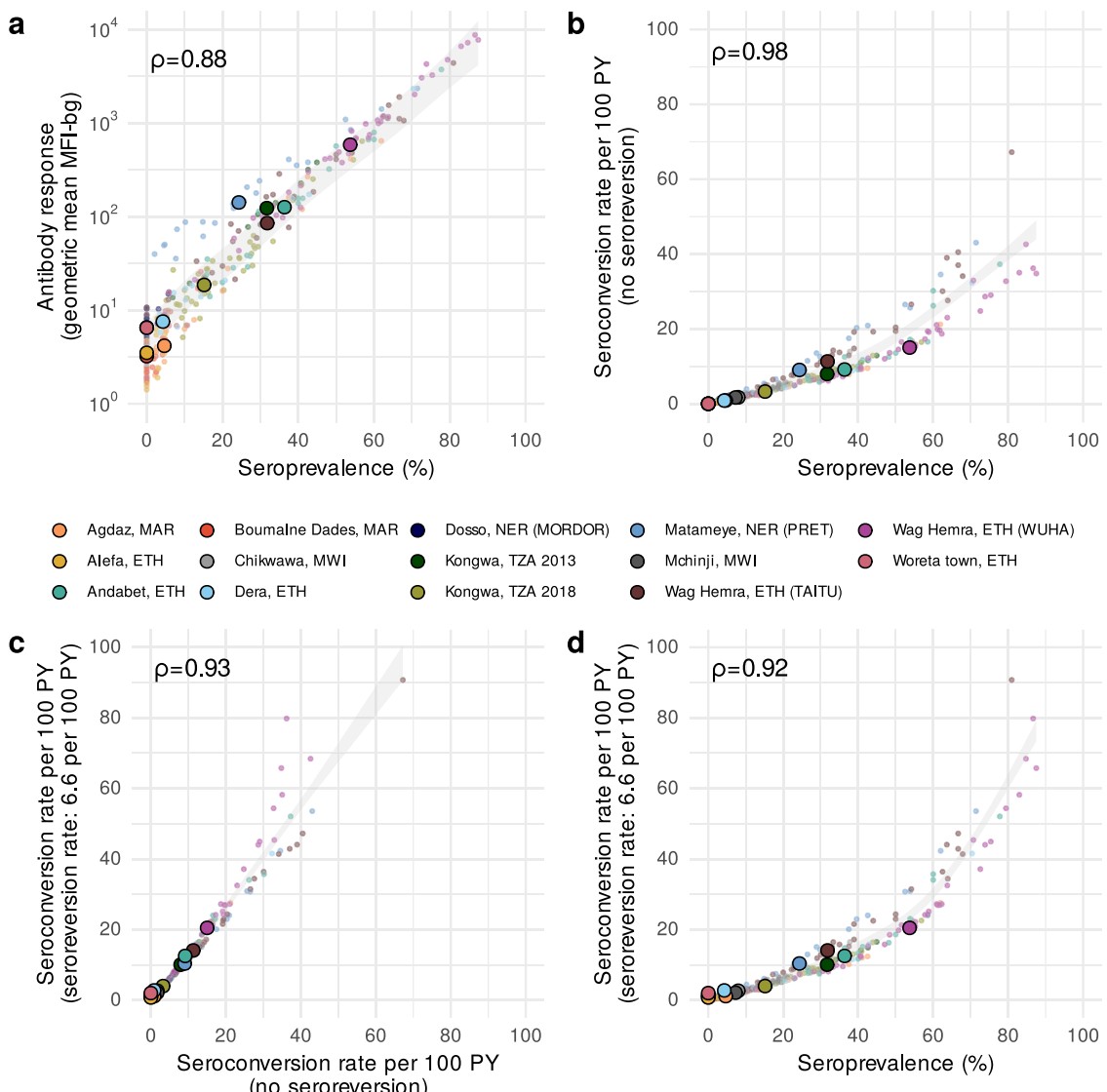

**Fig. 3 | Relationship between different cluster-level serologic summary measures. a** Geometric mean IgG response versus seroprevalence. **b** Seroconversion rate versus seroprevalence. **c** Seroconversion rate allowing for seroreversion versus a seroconversion rate that assumes no seroreversion. **d** Seroconversion rate that assumes seroreversion versus seroprevalence. Spearman rank correlations across sampling clusters are shown for each comparison. In all panels, small points represent cluster-level estimates and medians for each study population are represented by larger points with black outline. Shaded bands show simultaneous 95% confidence intervals for a spline fit through cluster-level estimates. A fixed seroreversion rate of 6.6 per 100 person-years was assumed in the catalytic model allowing for seroreversion (Methods). Figure 1 includes each study's sample size (median cluster size = 40 children). *PY* person-years, *ETH* Ethiopia, *MAR* Morocco, *MWI* Malawi, *NER* Niger, *TZA* Tanzania. Source data are provided with this paper. Created with notebook https://osf.io/wcv86.

model), a reversible catalytic model allowing for seroreversion (corresponding to a susceptible-infected-susceptible, SIS, model), and a semiparametric hazard model that allowed for age-varying seroconversion rates.

A key question for elimination programs is whether the additional complexity in analysis leads to more useful information for identifying populations with ongoing ocular *C. trachomatis* transmission. We estimated each summary at the sampling cluster and population levels across the 14 study populations and found high correlation between all measures (Spearman $\rho \geq 0.88$ for all comparisons, Fig. 3). Cluster-level seroprevalence was strongly, linearly related with geometric mean IgG levels on the $\log_{10}$ scale (Fig. 3a), and there was a strong, non-linear relationship between seroprevalence and the seroconversion rate (Fig. 3b). Catalytic models that allowed for seroreversion shifted seroconversion rates up, particularly for clusters with a seroconversion rate greater than 20 per 100 person-years but had minimal influence on estimates at intermediate and lower levels of transmission (Fig. 3c).

Similarly, seroprevalence was tightly linked with seroconversion rates that allowed for seroreversion in settings with lower prevalence, and greater dispersion was observed in hyperendemic populations (Fig. 3d). The most complex model, a semiparametric proportional hazards spline model that allowed for an age-varying seroconversion rate, required more data than would be typically available at the sampling cluster level and thus could only be estimated at the population level where its estimates aligned closely with a simpler, constant rate model (Supplementary Fig. 3). These results show that all serologic summary measures provide similar information when averaged at the cluster or population level across all ranges of transmission and particularly at lower levels of transmission.

## Comparing trachoma indicators in the presence and absence of MDA

As most trachoma program interventions act to clear infection or reduce its transmission, we sought summaries that could capture

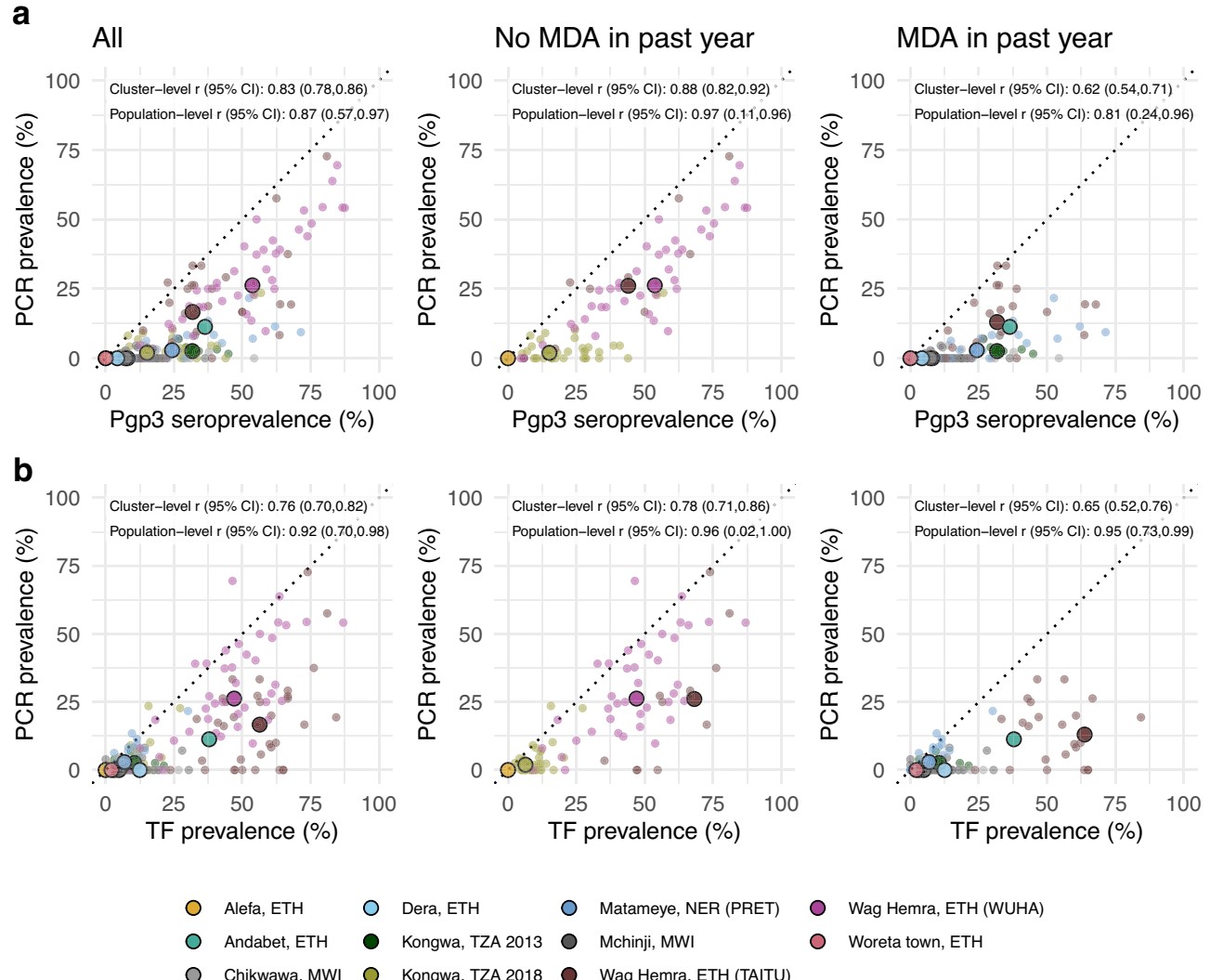

**Fig. 4 | Relationship between trachoma biomarkers in the presence and absence of recent mass distribution of azithromycin (MDA). a** Correlations between cluster PCR prevalence and seroprevalence overall and stratified by whether the study population had received MDA in the previous year. **b** Correlations between PCR prevalence and TF prevalence overall and stratified by whether the study population had received MDA in the previous year. In all panels, medians across clusters for each study population are represented by larger points with black outline. Each plot includes the identity line (dotted) and Pearson correlations at the cluster- and population levels. 95% confidence intervals (CIs) are based on 1000 bootstrapped samples, holding populations fixed and resampling clusters with replacement. Population-level estimates are included for Andabet, Dera, Woreta town, and Alefa, Ethiopia, but cluster-level PCR prevalence was not available for these populations. Figure 1 includes each study's sample size. *ETH* Ethiopia, *MWI* Malawi, *NER* Niger, *TZA* Tanzania, *TF* trachomatous inflammation– follicular. Source data are provided with this paper. Created with notebook https:// osf.io/rt825.

variation in populations' *C. trachomatis* infection burden. Seroprevalence and PCR prevalence aligned closely at the population level (*r* = 0.87; 95% CI: 0.57–0.97, Fig. 4a). At both the cluster- and population levels, seroprevalence was almost always greater than or equal to PCR prevalence, underscoring the sensitivity of seroprevalence as a measure of current or past infection. However, PCR prevalence was highly variable across levels of seroprevalence; low PCR prevalence in the context of high seroprevalence may imply previously high levels of infection that had been controlled by MDA or other interventions. Relationships between TF and PCR prevalence were similar (population-level *r* = 0.92; 95% CI: 0.70–0.98, Fig. 4b).

As shown in prior work[41], TF and seroprevalence were more strongly correlated with PCR prevalence in the absence of MDA in the past year (Fig. 4a, b). In the absence of MDA, the relationship between seroprevalence and PCR prevalence was nearly linear, particularly in high transmission settings. In the presence of recent MDA, correlations between trachoma indicators weakened presumably because IgG and

TF prevalence remain elevated as durable indicators of trachoma even after infections have been reduced through MDA.

We also stratified the analyses by child age and observed similar correlations between indicators among children aged 1–9 years compared to children aged 1–5 years (Fig. 5). When summarizing IgG responses as seroconversion rate instead of seroprevalence, overall results and subgroup results by recent MDA and age group were similar (Supplementary Fig. 4), reflecting the tight correlation between serologic summaries (Fig. 3). Our results show that serologic summaries align with PCR prevalence as well or better than TF, the current programmatic indicator, and that serology performs well even among preschool-aged children. In populations that have recently undergone MDA, IgG, and TF will likely remain elevated even as infection prevalence falls.

**Assessing serologic thresholds for elimination of infections**
Due to the wide variability in trachoma indicators both within and between populations, we used a non-parametric approach to evaluate

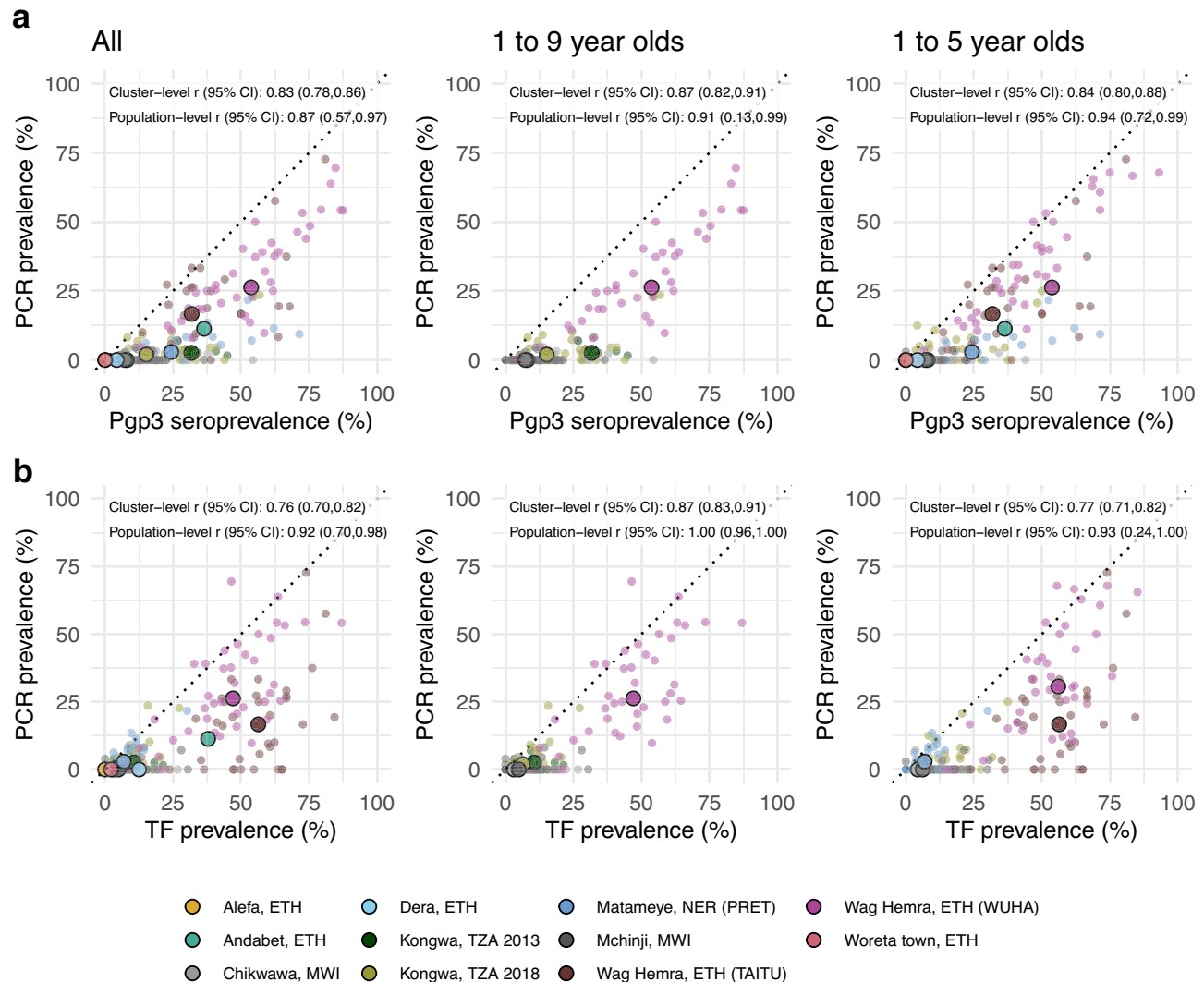

**Fig. 5 | Relationship between trachoma biomarkers in different monitoring age ranges. a** Correlations between cluster PCR prevalence and seroprevalence overall and stratified by age of monitoring population. **b** Correlations between *C. trachomatis* PCR prevalence and TF prevalence overall and stratified by age of monitoring population. In all panels, medians across clusters for each study population are represented by larger points with black outline. Each plot includes the identity line (dotted) and Pearson correlations at the cluster- and population levels. 95% confidence intervals (CIs) are based on 1000 bootstrapped samples, holding populations fixed and resampling clusters with replacement. Population-level estimates are included for Andabet, Dera, Woreta town, and Alefa, Ethiopia, but cluster-level PCR prevalence was not available for these populations. Figure 1 includes each study's sample size. *ETH* Ethiopia, *MWI* Malawi, *NER* Niger, *TZA* Tanzania, *TF* trachomatous inflammation−follicular. Source data are provided with this paper. Created with notebook https://osf.io/rt825.

potential serologic summary thresholds. Based on the alignment between seroprevalence and PCR at the cluster level and the relatively large number of clusters, we assessed sensitivity and specificity using cluster-level values. At each potential threshold, sensitivity and specificity were calculated across the 281 clusters with PCR measurements, 112 of which had at least one infection detected by PCR. Although the presence of a single infection is a stringent threshold, an appropriate level of PCR prevalence required to prevent blindness due to trachoma has yet to be established. Under this definition, a threshold with high sensitivity would result in more infected clusters being detected, while a threshold with high specificity might imply less unnecessary intervention. While overtreatment should be avoided, we focused on thresholds with high sensitivity with the goal of elimination in mind.

There was good classification of clusters with and without *C. trachomatis* infections based on either seroprevalence or seroconversion rates (AUC = 0.92). Thresholds of 13.5% seroprevalence (Fig. 6) or 2.75 seroconversions per 100 person-years (Supplementary Fig. 5, Supplementary Data 2) had 90% sensitivity and 69–75% specificity to

identify clusters with any PCR-detected infections. Thresholds of 22.5% seroprevalence and 5.75 seroconversions per 100 person-years had 80% sensitivity and 92–93% specificity. Thresholds were conservative when compared against population-specific sensitivity estimates in high transmission settings, but sensitivity dropped quickly in lower transmission populations with few infected clusters (*n* = 4 in Chikwawa, Malawi, and *n* = 6 in Mchinji, Malawi) (Fig. 6a).

In a sensitivity analysis we repeated the approach including only children aged 1–5 years, since restricting to younger ages should result in overall lower seroprevalence and potentially different seroconversion rates. We found the classification of clusters with and without *C. trachomatis* infections were almost identical to the main analysis (AUC = 0.92), and thresholds of 12.5% seroprevalence and 2.75 conversions per 100 person-years had 90% sensitivity and 69–71% specificity to identify clusters with any PCR-detected infections (Supplementary Fig. 6, Supplementary Data 3, Supplementary Data 4).

In a supplementary assessment of the thresholds, we applied them to populations with very low trachoma prevalence that were not

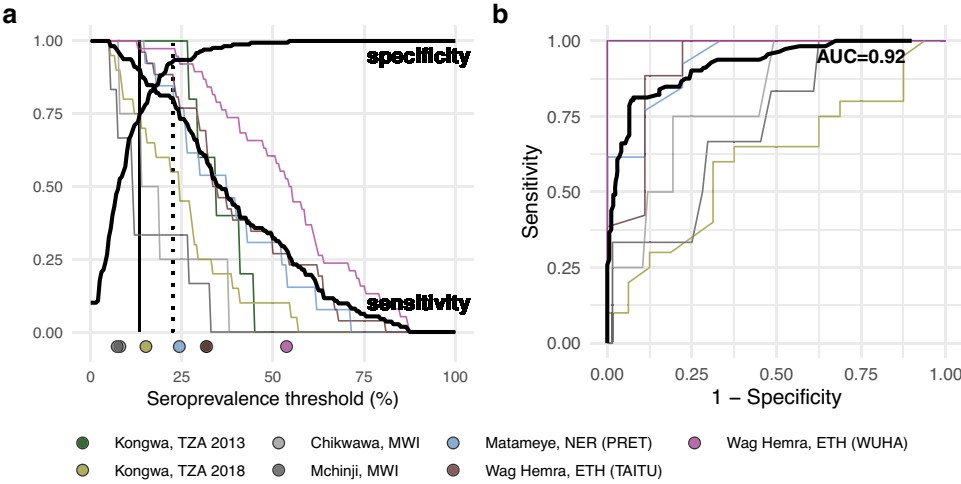

**Fig. 6 | Identification of clusters with *C. trachomatis* infection using seroprevalence. a** Sensitivity and specificity curves for identification of clusters with any *C. trachomatis* PCR infections based on different seroprevalence thresholds among children 1–9 years old (*n* = 281 clusters, 112 clusters with PCR > 0%). Vertical lines delineate sensitivity at 90% (solid line, 13.5% seroprevalence) and 80% (dotted line, 22.5% seroprevalence). Thin lines show sensitivity for each study population with cluster-level PCR measurements. Points at the bottom of the panel mark seroprevalence estimates for each study population. **b** Receiver operating

characteristic (ROC) curves for all clusters (thick black line) and for each study population (thin colored lines) with overall area-under-the-curve (AUC). Note that the ROC curve for Kongwa, TZA 2013 lies beneath the curve for Wag Hemra, ETH (WUHA). Figure 1 includes each study's sample size. *ETH* Ethiopia, *MWI* Malawi, *NER* Niger, *TZA* Tanzania. Supplementary Data 1 includes underlying sensitivity and specificity values at each cutoff. Source data are provided with this paper. Created with notebook https://osf.io/vgekh.

included in the threshold analysis because cluster-level PCR results were unavailable (Dosso, Alefa, and Woreta town, see Figs. 1 and 2). District-level PCR from Alefa and Woreta town, Ethiopia were estimated to be 0%, and a recent study from the same population in Dosso, Niger found no *C. trachomatis* in pooled metagenomic testing of ocular swabs[38]. In these three populations, 98% of clusters (60/61) fell below the serologic thresholds for 90% sensitivity, supporting the interpretation of low levels of infection below the estimated thresholds.

## Discussion

Using antibody measurements from 14 populations across a gradient of *C. trachomatis* transmission intensity, we demonstrated that IgG Pgp3 age-seroprevalence curves flatten and seroprevalence and seroconversion rates progress toward zero as populations approach elimination, consistent with biological assumptions and observations across other pathogens[40,42]. To our knowledge, this is the largest synthesis of *C. trachomatis* serology and infection data spanning globally relevant transmission settings from hyper-endemic to elimination. Summaries of IgG responses to *C. trachomatis* antigen Pgp3 among children aged 1–9 years aligned similarly with PCR prevalence and with one another at both the cluster- and population levels, indicating that cluster-level Pgp3 summaries reflect variation in *C. trachomatis* infections and ongoing transmission. Combining information across all populations, we found that thresholds of 13.5% seroprevalence and 2.75 seroconversions per 100 person-years identified clusters with any infection with high sensitivity (>90%) and moderate specificity (69 to 75%). We observed similar results when analyses were restricted to children aged 1–5 years, an age group that is potentially easier to sample in household surveys as they are not yet in school.

We demonstrated that seroprevalence and seroconversion rates perform as well or better as correlates of population levels of infection compared with clinical TF, and serology is arguably more objective, granular, and scalable for surveillance. Although clinical signs are important markers of disease progression, MDA is targeted towards *C. trachomatis* infections—if clinical signs continue to be present in the absence of antibody responses, further investigation may be

warranted to determine other potential causes of inflammation. Compared to PCR, which also measures *C. trachomatis* infections, serology is cost-effective, with simpler collection procedures, the opportunity to leverage banked samples, and measures exposure over time versus at a single point, an advantage in low prevalence settings.

By focusing on parsimonious, generalizable methods, our aim was to present seroepidemiologic approaches that could be easily applied and interpreted by trachoma control programs. Trachoma programs currently rely on district-level estimates of active trachoma, and this analysis may be extended to develop district-level thresholds as an increasing number of district-level, population-based serologic surveys become available. Cluster-level estimates, as investigated here, may be a promising alternative to identify focal areas for intervention, particularly in combination with geostatistical methods to improve precision and project across unsurveyed regions[43].

Programmatic decisions related to suspension or reinitiation of antibiotic MDA primarily take place in low to intermediate transmission settings, and this study demonstrates that the information serologic surveillance provides in such settings should be robust to choice of summary statistic. Serologic summaries of increasing complexity were tightly correlated with one another, and the relationship was nearly linear in settings with low to intermediate transmission (Fig. 3). Quantitative IgG responses potentially provide information at very low levels of *C. trachomatis* transmission when most children are seronegative[40], but a challenge is that IgG responses are in arbitrary units that may not be directly comparable across studies. Members of our team recently developed a chimeric, monoclonal antibody to help facilitate direct comparison of quantitative antibody levels across studies[18]. Seroprevalence and seroconversion rates are standard transformations of the quantitative IgG response that are easier to compare across studies and labs. A potential drawback of seroprevalence and seroconversion is that they may be sensitive to choice of the seropositivity cutoff, potentially affecting programmatic decisions based on thresholds. In a sensitivity analysis, we found that identification of clusters with *C. trachomatis* infection based on seroprevalence and seroconversion rates were robust to large changes in the seropositivity cutoff (Supplementary Fig. 7). The seroconversion rate is tightly linked to the slope of the age-seroprevalence curve

(details in Methods). Additionally, the seroconversion rate implicitly adjusts for age (not guaranteed with mean IgG or seroprevalence) and can be interpreted as a measure of a pathogen's force of infection. Although we did not consider complicated seroconversion models that allowed for changes in transmission over time, such complexity may be less relevant when analyzing measurements from young children because their antibody responses reflect recent infections[44–46].

Our results support estimation of age-seroprevalence curves among young children combined with estimation of seroconversion rates using a simple, single-rate model to characterize trachoma transmission. Our findings demonstrate that the slope of the age-seroprevalence curve and the seroconversion rate are sensitive and moderately specific markers of population-level *C. trachomatis* infection. Seroprevalence is another simple, robust summary measure that aligned with infections as well as seroconversion rate, particularly in the absence of recent MDA, but care should be taken to ensure similar age structure between populations to avoid bias.

We found that serologic summaries were sensitive indicators of *C. trachomatis* infection but have only moderate specificity: antibody responses were nearly always present in populations with infections, but the IgG signal persisted even if recent MDA reduced infection to low levels. This pattern suggests that populations requiring treatment will rarely be missed by serology but further verification with PCR could prevent overtreatment in the case of false positives. Because IgG responses to Pgp3 are durable, age-seroprevalence curves and statistics estimated from them may reflect historical transmission patterns and remain high despite low levels of infection resulting from control measures or secular trends. Restricting analysis to younger children (e.g., children born after discontinuing antibiotic MDA) or carefully accounting for past MDA treatments in the interpretation of antibody data may decrease false positives detected by IgG.

Here, we focused on IgG among young children and its relationship with *C. trachomatis* infection to extend a previous study that combined nine trachoma serology surveys and compared IgG seroconversion rates with clinical indicators among people of all ages[44]. Limiting the analysis to young children rather than all ages enabled us to simplify the modeling approach used to estimate seroconversion rates under the assumption that transmission was approximately stable over the age range. We focused on the relationship between serology and PCR infection to assess the utility of serology to monitor differences in pathogen transmission.

The threshold analyses show that Pgp3 serology provides good overall discrimination of the presence of *C. trachomatis* infections in a community (AUC = 0.92), even when limited to children 1–5 years old, but there are two caveats to the generalizability of the thresholds estimated in this analysis. First, we used clusters as the unit of analysis —a much finer spatial scale than the current scale of decision-making for elimination programs, which is typically at the administrative district level. We felt a similar, non-parametric threshold analysis at the district (study) level would not be informative with only 11 studies that included *C. trachomatis* infection measurements (Fig. 1), versus 281 unique clusters (Fig. 6). Yet, the small number of children in any single cluster (median = 40) means that rare infections could be missed without a larger sample size as populations approach elimination (for example, if true *C. trachomatis* prevalence were 1%, then $n = 300$ children would need to be tested to have a 95% probability of detecting at least one *C. trachomatis* infection, assuming perfect test sensitivity and independent tests, where $1 - [1 - P(Ct^+)]^n \geq 0.95$. It would be rare to encounter 300 young children in a single sampling cluster, so averaging over a larger spatial scale or over repeated surveys in the same clusters could improve the robustness of *C. trachomatis* infections as a gold standard against which serology is compared. A second caveat is that the analysis included the full range of transmission settings, and lower thresholds may be needed to best discriminate in low transmission settings (similar to Malawi or Kongwa 2018 populations,

Fig. 6). Building from these promising results, in future work our team plans to assess serology thresholds at the district-level with a focus on near-elimination and post-elimination settings.

This work is subject to limitations. First, the data we analyzed were compiled from different previously published studies and sites that were available to us. Our results may not be representative of all populations with ongoing trachoma transmission. For example, the included studies come from five countries in Africa, but trachoma continues to be endemic in the other regions of Africa, Latin America, Asia, and the Pacific Islands[47]. As a secondary analysis, our work was also affected by design features of individual studies, such as site selection in RCTs, which were typically not selected using probability samples. Second, all models used to estimate seroconversion, except for the semiparametric spline model, assumed homogeneity of seroconversion rate over age and time. For most studies, we expected stable transmission among 1–9-year-olds in the recent past. Our analysis used an estimate of seroreversion from one longitudinal cohort[48], but data beyond this cohort is limited and seroreversion may be lower in higher transmission settings[49]. In a sensitivity analysis no single seroreversion value resulted in the best model fit across populations (Supplementary Fig. 8). Finally, we were unable to evaluate more specific subgroups, such as populations with different MDA histories or populations experiencing recrudescence, due to limited data. Based on the variability observed between populations, refinement of thresholds may be warranted as more data are collected.

In conclusion, IgG responses to *C. trachomatis* antigen Pgp3 among 1–9-year-olds aligned closely with ongoing transmission as measured by PCR and TF across a range of trachoma endemicity settings. Consistent with patterns observed across other pathogens, age-dependent Pgp3 seroprevalence curves became flatter and seroconversion rates declined to zero as populations approached elimination—providing a coherent framework for monitoring *C. trachomatis* transmission in seroepidemiologic studies. Serologic summaries estimated via a range of model complexity were sensitive markers of ocular chlamydial infection and were consistent with one another, implying that simpler approaches are likely sufficient to capture variation in transmission. These results support the use of serologic surveys to inform trachoma programs as populations approach and achieve elimination.

## Methods
### Contributing studies
We gathered data from published trachoma serology surveys, with an emphasis on IgG antibody responses to Pgp3 collected among children aged 1–9 years and relatively recent reports (Supplementary Table 1). All studies were conducted between 2012 and 2019. Children below the age of 1 year were excluded to mitigate the influence of maternal antibodies. Based on the original study designs, measurements in Matameye, Niger (PRET) and Wag Hemra, Ethiopia (TAITU) were restricted to children aged under 6 years; in Dosso, Niger (MORDOR), the study was limited to children aged under 5 years; and in Alefa, Andabet, Dera, and Woreta town, Ethiopia, infections were measured among children aged 1–5-years, while serology and TF were measured among children aged 1–9-years. For measurements collected as part of a population-based surveys, multi-level cluster random sampling was used. For measurements collected as part of a randomized controlled trial, we combined data across arms unless otherwise noted (e.g., stratification by recent MDA). Due to changing transmission and/or control interventions in studies with repeated cross-sectional data (WUHA in Wag Hemra, Ethiopia; Kongwa, Tanzania from 2012-2015; MORDOR in Dosso, Niger), we included only the most recent year of measurements with serology, and PCR measurements if available, for each population. We excluded clusters with fewer than 15 children measured to ensure sufficient information to estimate cluster-level means ($n = 22$ clusters excluded, reduced from 481 to 459). Measurement sample sizes reflect distinct samples from individual children.

Details of serologic, clinical, and PCR measurements can be found in the published reports for each study[26–38]. In Malawi, dried blood spots were tested for IgG antibodies against Pgp3 using ELISA. Samples were added to Immulon 2HB plates (Southern Biotech, Birmingham, AL, USA) pre-sensitized with Pgp3 protein and incubated for 2 hours. After 4 washes, plates were incubated with anti-human IgG-HRP (Southern Biotech) for 1 hour. Plates were then washed and incubated with 3,3′,5,5′-tetramethylbenzidine (KPL, Gaithersburg, MD, USA). The reaction was stopped with 1 N H2SO4 and plates were read at 450 nm on a microplate reader. The seropositivity cutoff was determined based on a finite mixture model[31]. ELISA-based population-level measurements of Pgp3 have been shown to have good agreement with multiplex bead assays[50].

In all other included studies, dried blood spots were analyzed for IgG antibodies to Pgp3 using a multiplex bead assay on a Luminex platform. Briefly, Pgp3-coupled beads were incubated with diluted sample for 1.5 hours, washed, and then incubated with anti-human IgG (Southern Biotech, Birmingham, AL, USA) and anti-human IgG4 (Southern Biotech) for 45 minutes. After additional washes, beads were incubated with phycoerythrin-labeled streptavidin (Invitrogen, Waltham, MA, USA) for 30 minutes, washed, and then incubated with phosphate-buffered saline (PBS) containing 0.5% BSA, 0.05% Tween-20 and 0.02% sodium azide. After a final wash, beads were resuspended in 1× PBS and read on a Bio-Plex 200 instrument (Bio-Rad, Hercules, CA). IgG levels were reported as median fluorescence intensity minus background (MFI-bg), and seropositivity cutoffs were generated using receiver operator characteristic (ROC) methods[14]. We used MFI-bg cutoffs defined by each study to assign seropositivity status. The MFI-bg cutoffs can vary slightly by study due to differences in antigen-bead coupling efficiency in different bead sets, and ranged from 882 (Kongwa, Tanzania 2013) to 1771 (Kongwa, Tanzania 2018), with a median of 1558 (Alefa, Andabet, Dera, and Woreta town, Ethiopia 2017). De-identified Source data provided with this study include the cutoffs used for each bead set. To assess robustness to seropositivity cutoffs, we compared seroprevalence and seroconversion rate calculated across a range of arbitrary cutoffs for four study populations at different levels of trachoma prevalence (Supplementary Fig. 7), which illustrate that the range over which cutoffs vary are unlikely to have a major influence on seroprevalence or seroconversion rate estimates.

Clinical disease was assessed by trained field graders according to the WHO simplified grading system[51], which defines TF as the presence of five or more follicles which are (a) at least 0.5 mm in diameter and (b) located in the central part of the upper tarsal conjunctiva. Conjunctival swabs were assessed for *C. trachomatis* DNA using PCR. In five out of seven studies with PCR measurements, cluster-level prevalence was estimated from individual-level results[29–31,35,36]. In one study (PRET), cluster-level prevalence was estimated from pooled results using maximum likelihood methods[26,52]. For Andabet, Dera, Woreta town, and Alefa, Ethiopia, *C. trachomatis* infection prevalence was estimated at the district level from pooled results using maximum likelihood methods[28].

Cluster-level prevalence was calculated as the number of children who were sero-, PCR- or TF-positive divided by the number of children tested for the respective outcome. When possible, "populations" were defined by districts or evaluation units currently used for trachoma monitoring. Population-level prevalence estimates were defined as the median value across clusters. Age-seroprevalence curves were estimated by pooling all measurements at the population level, calculating seroprevalence for one-year age groups, and fitting cubic spline models to generate smooth trend lines.

### Estimating serology-based summary measures

We summarized IgG responses as geometric mean IgG levels, seroprevalence, and seroconversion rates from several models. The seroconversion rate from a current status, single-rate catalytic model assuming no seroreversion is closely tied to the slope of age-seroprevalence curve[53]. Specifically, the hazard (seroconversion rate) is equal to the slope of the age-seroprevalence curve divided by the complement of the seroprevalence at age $A = a$. It can be shown that the seroconversion rate based on this model can be estimated as the exponentiated intercept from a generalized linear model with binomial error structure and a complementary log-log link[54]:

$$\log[-\log(1 - P(Y = 1|A))] = \log(\lambda) + \log(A) \qquad (1)$$

where $Y$ represents individual-level serostatus (1: seropositive, 0: seronegative or equivocal), $A$ is the child's age in years, and $\lambda$ is the seroconversion rate. The seroconversion rate provides an estimate of the force of infection, an epidemiological parameter that denotes the rate at which susceptible individuals in the population become infected.

Next, we extended this model to allow for seroreversion. Cross-sectional data do not contain sufficient information to reliably estimate both seroreversion and seroconversion rates[55]. Therefore, we fit a binomial maximum likelihood model for seroconversion with fixed seroreversion rates ranging from 0.02 to 0.20 per person-year[22,44]:

$$P(Y = 1|A) = \frac{\lambda}{\lambda + \rho}[1 - \exp(-(\lambda + \rho) \times A)] \qquad (2)$$

where ρ is the assumed seroreversion rate. In our main analyses, we assumed a seroreversion rate of 6.6 per 100 person-years based on a longitudinal cohort in Kongwa, Tanzania monitored in the absence of MDA[48], which is likely conservative for higher transmission settings[49]. No level of seroreversion consistently provided the best fit across populations (Supplementary Fig. 8). This model did not converge in several clusters (6 out of 416 clusters with seroprevalence >0%, or 1.4%), which often contained few seropositive children.

Finally, we fit semiparametric models to allow seroconversion to vary by age[53]. We fit semiparametric models only at the population level, as they require more data than are typically available at the sampling cluster level (around 20–40 children). We fit a generalized linear model with complementary log–log link for seropositivity:

$$\log[-\log(1 - P(Y = 1|A))] = g(A) \qquad (3)$$

where *g(A)* is a flexible function for age modeled using cubic splines. To estimate the average seroconversion rate in each population, $\lambda$, we integrated over age using predictions from the model. We used the relationship between the hazard $\lambda(a)$ and cumulative incidence $F(a)$, where $F(a)$ is the predicted seroprevalence at age $a$ (details in ref. 42).

$$\lambda = \int_{a_1}^{a_2} \lambda(a)da = \frac{\log[1 - F(a_1)] - \log[1 - F(a_2)]}{a_2 - a_1} \qquad (4)$$

### Comparison of different trachoma indicators

We used the Pearson correlation coefficient to compare serologic summaries and TF prevalence with PCR prevalence at the population and cluster levels. We conducted this analysis across all populations with PCR measurements and explored the impact of contextual factors by stratifying clusters into subgroups defined by MDA in the past year (yes/no) and age range (1–9-year-olds/1–5-year-olds). We estimated asymptotic 95% confidence intervals for study-level correlation using Fisher's Z transform. To estimate 95% confidence intervals for cluster level correlation estimates we used a non-parametric bootstrap (1000 replicates) that resampled clusters with replacement, stratified by study.

We used the Spearman rank correlation coefficient to compare different serologic summaries with one another at the cluster level,

allowing for non-linearity in the relationships. We estimated simultaneous confidence intervals from a cubic spline to summarize the relationship between markers at the cluster level, including a random effect for study to allow for correlated measures within-study.

## Serologic thresholds for elimination

We used a non-parametric approach to assess thresholds of seroprevalence or seroconversion rates to classify clusters without PCR infections. For overall estimates, we combined clusters from all populations with PCR measurements, for a total of 281 clusters and 112 clusters with at least one PCR-detected infection. We defined a range of cutoffs covering nearly all cluster-level estimates (0 to 100% for seroprevalence, 0 to 50 per 100 person-years for seroconversion rate) and divided the range into 200 increments. At each step, we calculated sensitivity as the proportion of clusters with serology values above the threshold among clusters with any infection, and specificity as the proportion of clusters with serology value less than or equal to the threshold among clusters with zero infections detected. We identified the highest thresholds achieving at least 90% and 80% sensitivity. We also presented these results as receiver operating characteristic (ROC) curves and calculated overall area-under-the-curve (AUC). We calculated sensitivity, specificity, and the ROC curve for each population separately.

## Ethics and inclusion

The secondary analysis protocol was reviewed and approved by the Institutional Review Board at the University of California, San Francisco (Protocol #20-33198). All primary data that contributed to the analysis were collected after obtaining informed consent from all participants or their guardians under separate, local human subjects research protocols in accordance with the Declaration of Helsinki. Members from each contributing primary research study have participated as collaborators and co-authors on the present analyses from their initial stages, including the design, interpretation, and summary of results. Co-authors were nominated by each study's principal investigator to represent the country and study teams that originally contributed the data. De-identified datasets made public through this analysis have been reviewed and approved by representatives from each study and conform with ethical guidelines set forth in the original protocols. Analyses were led by investigators at the University of California, San Francisco with guidance and input from all co-authors to incorporate local stakeholder perspectives.

## Reporting summary

Further information on research design is available in the Nature Portfolio Reporting Summary linked to this article.

# Data availability

De-identified data to reproduce this work are publicly available without restriction in the Open Science Framework repository, https://osf.io/e6j5a/[56]. Processed, de-identified data generated in this study are additionally provided in the Supplementary Information/Source Data file. Source data are provided with this paper.

# Code availability

R version 4.2.2 (2022-10-21, "Innocent and Trusting") was used for this analysis[57]. Open source code and instructions to reproduce all analyses are available in the Open Science Framework repository, https://osf.io/e6j5a/[56].

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

## Acknowledgements

This work was supported by the National Institute of Allergy and Infectious Diseases (R01-AI158884 to B.F.A.). The findings and conclusions in this article are those of the authors and do not necessarily represent the official position of the National Institutes of Health or the Centers for Disease Control and Prevention. Use of trade names is for identification only and does not imply endorsement by the Public Health Service or by the U.S. Department of Health and Human Services. The authors alone are responsible for the views expressed in this article, and they do not necessarily represent the views, decisions, or policies of the institutions with which they are affiliated. A.W.S. is a staff member of the World Health Organization.

## Author contributions

Following CRediT taxonomy: conceptualization (C.T., A.W.S., D.L.M., S.D.N., P.J.L., B.F.A.), data curation (C.T., D.L.M., B.F.A.), formal analysis (C.T., S.C., B.F.A.), funding acquisition (B.F.A., D.L.M.), investigation (C.T., B.F.A.), methodology (C.T., B.F.A.), project administration (D.L.M., B.F.A.),

resources (D.L.M., S.D.N., J.D.K., T.M.L., A.A., S.A., A.M.A., E.K.C., A.A.D., E.B.G., S.G., J.H., B.K., K.K., R.M., B.N., F.S., Z.T., S.K.W., T.Z.), software (C.T., B.F.A.), supervision (A.W.S., D.L.M., S.D.N., S.K.W., T.M.L., J.D.K., P.J.L.), validation (C.T., B.G., D.L.M., B.F.A.), visualization (C.T., B.F.A.), writing—original draft preparation (C.T., B.F.A.), writing—review & editing (C.T., A.W.S., D.L.M., S.D.N., J.D.K., T.M.L., P.J.L., K.A., A.A., S.A., A.M.A., E.K.C., S.C., A.A.D., E.B.G., S.G., J.H., B.K., K.K., R.M., B.N., F.S., Z.T., S.K.W., D.M.W., T.Z., B.F.A.).

## Competing interests

The authors declare no competing interests.

## Additional information

[1]Francis I. Proctor Foundation, University of California San Francisco, San Francisco, CA 94158, USA. [2]Global Neglected Tropical Diseases Programme, World Health Organization, Geneva, Switzerland. [3]Division of Parasitic Diseases and Malaria, Centers for Disease Control and Prevention, Atlanta, GA 30329, USA. [4]The Carter Center, Atlanta, GA 30307, USA. [5]Department of Ophthalmology, University of California San Francisco, San Francisco, CA 94158, USA. [6]Institute for Global Health Sciences, University of California San Francisco, San Francisco, CA 94143, USA. [7]Department of Epidemiology and Biostatistics, University of California San Francisco, San Francisco, CA 94143, USA. [8]Neglected Tropical Diseases Support Center, Task Force for Global Health, Atlanta, GA 30030, USA. [9]Division of Epidemiology, Department of Public Health Sciences, University of California Davis School of Medicine, Davis, CA, USA. [10]Programme National de Santé Oculaire, Niamey, Niger. [11]Programme National de Lutte Contre la Cecité, Niamey, Niger. [12]The Carter Center Ethiopia, Addis Ababa, Ethiopia. [13]Infection Biology, Addis Ababa University, Addis Ababa, Ethiopia. [14]The Carter Center Niger, Niamey, Niger. [15]Amhara Regional Health Bureau, Bahir-Dar, Ethiopia. [16]Service of Ocular and Otological Diseases, Epidemiology and Disease Control Directorate, Ministry of Health, Rabat, Morocco. [17]Blantyre Institute for Community Outreach, Blantyre, Malawi. [18]Federal Ministry of Health, Addis Ababa, Ethiopia. [19]Johns Hopkins School of Medicine, Dana Center for Preventive Ophthalmology, Wilmer Eye Institute, Baltimore, MD, USA. [20]Amhara Public Health Institute, Bahir-Dar, Ethiopia. ✉e-mail: ben.arnold@ucsf.edu

