## [Peer Review File · Nature Communications]

Monitoring transmission intensity of trachoma with serologyReviewers' Comments:

Reviewer #1:

Remarks to the Author:

This is an excellent investigation of the use of serology for surveillance of trachoma led by an accomplished team at the Proctor Foundation. A large amount of high quality data is analysed, and there are a number of practical applications which may greatly aid efforts to document the elimination of trachoma in Africa.

I have no major comments, and a number of minor comments:

- In the abstract, please specify how infection was identified.

"A seroprevalence threshold of 13.5% (seroconversion rate 2.75 per 100 person-years) identifies clusters with any infection at high sensitivity (>90%) and moderate specificity (69-75%)."

- Can the authors please provide a summary of the lab methods used for the serological assay in the methods section. Presumably the same assay was used for all studies.

- Methods (line 409 – 410)

"We used cutoffs defined by each study to assign seropositivity status."

So a different cutoff was used for each study – how much did these vary in terms of MFI?

- A personal opinion is that I don't think the serological data presented in Figure 3 provides much value. All values are essentially transformations of the underlying IgG MFI data, hence the excellent agreement. In my view, this data would be better as a Supplementary Figure. The authors should feel to disagree. The really interesting associations are those shown in Figures 4 and 5.

- I think the classification analysis of clusters presented in Figure 6 is very clever – and useful. The following comments reflect my enthusiasm for the idea, so addressing them all may well go beyond the scope of this paper.

Having a sero-prevalence threshold that is consistent with trachoma elimination is a very useful thing. However, the selection of this cut-off surely varies across multiple dimensions: transmission intensity, age distribution, number of samples, etc. It would be interesting to see additional analyses to investigate whether this cutoff is universal or not.

An additional point is that the "gold standard" method is whether all PCR measurements test negative. This makes sense – however, it is very dependent on the number of samples. A survey with a larger number of negative samples provides stronger evidence of elimination.

Reviewer #2:

Remarks to the Author:

I like the manuscript and it is sufficiently novel for publication. It is written in a way which should be accessible to more policy (potentially less quantitatively) minded people, but the mathematical methods are still well described and the code for producing the results is included. I appreciate how the authors have shown a clear theoretical development, but also returned to a clear policy recommendation with an associated uncertainty. The limitations of the study (e.g., using data from different studies) are recognised. I expect the paper will be an important contribution, not just for Trachoma, but also other NTDs.

The authors should have the data available for the published version unless they have a justification

not to include it (I couldn't see it in the repository for the code). The authors should be consistent, using either 'serologic' or 'serological'.

NCOMMS-23-06650

Monitoring transmission intensity of trachoma with serology

RESPONSE TO REVIEWER COMMENTS

Reviewer #1 (Remarks to the Author):

(1)

This is an excellent investigation of the use of serology for surveillance of trachoma led by an accomplished team at the Proctor Foundation. A large amount of high quality data is analysed, and there are a number of practical applications which may greatly aid efforts to document the elimination of trachoma in Africa.

I have no major comments, and a number of minor comments:

Response: Thank you for these kind remarks and your thoughtful review. We have incorporated your feedback to improve the manuscript.

(2)

• In the abstract, please specify how infection was identified.

“A seroprevalence threshold of 13.5% (seroconversion rate 2.75 per 100 person-years) identifies clusters with any infection at high sensitivity (>90%) and moderate specificity (69-75%).”

Response: We clarified the sentence so that it includes how infection was identified, by PCR (changes underlined):

Lines 54-57

“A seroprevalence threshold of 13.5% (seroconversion rate 2.75 per 100 person-years) identifies clusters with any PCR-identified infection at high sensitivity (>90%) and moderate specificity (69-75%).”

(3)

• Can the authors please provide a summary of the lab methods used for the serological assay in the methods section. Presumably the same assay was used for all studies.

Response: We have added a synopsis of the laboratory methods. Of 14 studies included, 13 used a multiplex bead assay on the Luminex platform. One study used an ELISA assay. Below, we have underlined the additions from paragraphs 2 and 3 in the Methods.

Lines 445 - 469

“Details of serologic, clinical, and PCR measurements can be found in the published reports for each study. In Malawi, dried blood spots were tested for IgG antibodies against Pgp3 using ELISA. Samples were added to Immulon 2HB plates (Southern Biotech, Birmingham, AL, USA) pre-sensitized with Pgp3 protein and incubated for two hours. After 4 washes, plates were incubated with anti-human IgG-HRP (Southern Biotech) for 1 hour. Plates were then washed and incubated with 3,3',5,5'-tetramethylbenzidine (KPL, Gaithersburg, MD, USA). The reaction was stopped with 1N H2SO4 and plates were read at 450 nm on a microplate reader. The seropositivity cutoff was determined based on a finite mixture model. ELISA-based population-level measurements of Pgp3 have been shown to have good agreement with multiplex bead assays.

In all other included studies, dried blood spots were analyzed for IgG antibodies to Pgp3 using a multiplex bead assay on a Luminex platform. Briefly, Pgp3-coupled beads were incubated with diluted sample for 1.5 hours, washed, and then incubated with anti-human IgG (Southern Biotech, Birmingham, AL, USA) and anti-human IgG4 (Southern Biotech) for 45 minutes. After additional washes, beads were incubated with phycoerythrin-labeled streptavidin (Invitrogen, Waltham, MA, USA) for 30 minutes, washed, and then incubated with phosphate buffered saline (PBS) containing 0.5% BSA, 0.05% Tween-20 and 0.02% sodium azide. After a final wash, beads were resuspended in 1X PBS and read on a Bio-Plex 200 instrument (Bio-Rad, Hercules, CA). IgG levels were reported as median fluorescence intensity minus background (MFI-bg), and seropositivity cutoffs were generated using receiver operator characteristic (ROC) methods.

(4)

• **Methods (line 409 – 410)**

“We used cutoffs defined by each study to assign seropositivity status.”

So a different cutoff was used for each study – how much did these vary in terms of MFI?

Response: We have provided some additional information in the Methods regarding the MFI seropositivity cutoffs and also pointed readers to a complete inventory of values for the studies, which are available through the de-identified dataset. Please see underlined additions, below:

Lines 469-479

"We used MFI-bg cutoffs defined by each study to assign seropositivity status. The MFI-bg cutoffs can vary slightly by study due to differences in antigen-bead coupling efficiency in different bead sets, and ranged from 882 (Kongwa, Tanzania 2013) to 1771 (Kongwa, Tanzania 2018), with a median of 1558 (Alefa, Andabet, Dera, and Woreta town, Ethiopia 2017). De-identified data provided with this study include the cutoffs used for each bead set. To assess robustness to seropositivity cutoffs, we compared seroprevalence and seroconversion rate calculated across a range of arbitrary cutoffs for four study populations at different levels of trachoma prevalence (Supplementary Fig. 7), which illustrate that the range over which cutoffs vary are unlikely to have a major influence on seroprevalence or seroconversion rate estimates."

(5)

• **A personal opinion is that I don't think the serological data presented in Figure 3 provides much value. All values are essentially transformations of the underlying IgG MFI data, hence the excellent agreement. In my view, this data would be better as a Supplementary Figure. The authors should feel to disagree. The really interesting associations are those shown in Figures 4 and 5.**

Response: We can see the perspective of the reviewer regarding Figure 3 because the different summary statistics are translating the same information into different units, under different assumptions. However, we have been hard-pressed to find an example in the published literature that demonstrates this rank-preserving relationship between mean IgG levels, seroprevalence, and seroconversion rates (estimated with- and without assuming seroreversion) for any pathogen, let alone *C. trachomatis*. We have also found that for many stakeholders that might use this information, it is not necessarily intuitive that the relationships would be so linear at low levels of transmission (e.g., at levels of seroprevalence <20%). We would like to keep Figure 3 in the main text because we found that this result was important to convince many members of our own research team / consortium of the possibility that various serologic summaries provide similar information about relative transmission.

(6)

• **I think the classification analysis of clusters presented in Figure 6 is very clever – and useful. The following comments reflect my enthusiasm for the idea, so addressing them all may well go beyond the scope of this paper.**

Having a sero-prevalence threshold that is consistent with trachoma elimination is a very useful thing. However, the selection of this cut-off surely varies across multiple

dimensions: transmission intensity, age distribution, number of samples, etc. It would be interesting to see additional analyses to investigate whether this cutoff is universal or not.

An additional point is that the “gold standard” method is whether all PCR measurements test negative. This makes sense – however, it is very dependent on the number of samples. A survey with a larger number of negative samples provides stronger evidence of elimination.

Response: The reviewer has raised several good questions related to the elimination threshold analyses. The influence of transmission intensity and age distribution on the thresholds can be inferred from the existing data presented in the paper (details below). Regarding the sample size, we completely agree that the number of samples is an important determinant in this type of assessment, particularly as populations approach elimination and PCR-infections become very rare. Since the monitoring strategy for trachoma will be limited at the cluster level by the number of children available to sample, increasing the number of samples tested would most likely require either repeated sampling in the same clusters or averaging information over multiple clusters as is currently done for clinical signs of trachoma (averaged up to the district level).

After discussing with the authorship team, we felt that given these initial promising results, it would be useful to extend the analysis to examine thresholds at the district level and to focus on populations that are near elimination or have achieved elimination. However, to do so in a rigorous way would require inclusion of data from additional studies, particularly in the near- and post-elimination context. We have started the process of recruiting and adding multiple new studies of this type to the harmonized data but felt that such expansion of data and analysis would justify a different paper. For example, it will require new coauthors to represent additional countries and studies and it will take more than a few months to properly incorporate new data and collaborators into the research consortium.

We have therefore revised the Discussion to more clearly explain these points raised by the reviewer to help clarify the strengths and limitations of the cutoffs presented in the paper (new additions underlined).

Lines 357 - 385

“The threshold analyses show that Pqp3 serology provides good overall discrimination of the presence of Ct infections in a community (AUC=0.92), even when limited to children 1 to 5 years, but there are two caveats to the generalizability of the thresholds estimated in this analysis. First, we used

clusters as the unit of analysis — a much finer spatial scale than the current scale of decision making for elimination programs, which is typically at the administrative district-level. We felt a similar, non-parametric threshold analysis at the district (study) level would not be informative with only 11 studies that included Ct infection measurements (Fig. 1), versus 281 unique clusters (Fig. 6). Yet, the small number of children in any single cluster (median = 40) means that rare infections could be missed without a larger sample size as populations approach elimination (for example, if true Ct prevalence were 1%, then n=300 children would need to be tested to have a 95% probability of detecting at least one Ct infection, assuming perfect test sensitivity and independent tests, where $1 - [1 - P(Ct^+)]^n \geq 0.95$). It would be rare to encounter 300 young children in a single sampling cluster, so averaging over a larger spatial scale or over repeated surveys in the same clusters could improve the robustness of Ct infections as a gold standard against which serology is compared. A second caveat is that the analysis included the full range of transmission settings, and lower thresholds may be needed to best discriminate in low transmission settings (similar to Malawi or Kongwa 2018 populations, Fig. 6). Building from these promising results, in future work our team plans to assess serology thresholds at the district-level with a focus on near-elimination and post-elimination settings.

Reviewer #2 (Remarks to the Author):

(1)

I like the manuscript and it is sufficiently novel for publication. It is written in a way which should be accessible to more policy (potentially less quantitatively) minded people, but the mathematical methods are still well described and the code for producing the results is included. I appreciate how the authors have shown a clear theoretical development, but also returned to a clear policy recommendation with an associated uncertainty. The limitations of the study (e.g., using data from different studies) are recognised. I expect the paper will be an important contribution, not just for Trachoma, but also other NTDs.

Response: Thank you for this supportive review. We are looking forward to sharing this work with policymakers working on NTDs.

(2)

The authors should have the data available for the published version unless they have a justification not to include it (I couldn't see it in the repository for the code).

Response: In the time since the initial submission, all data contributors have agreed to make de-identified data public alongside the article. We have made all data publicly available in two locations. First, the base, harmonized datasets that are slightly more expansive than those used in the analysis (<https://osf.io/ykjc4/>), and the actual analysis datasets created for this paper (<https://osf.io/e6j5a/>). In tandem, we enhanced the reproducible workflow for this paper so that it reads-in datasets from their public locations. The code can now run on a standard desktop (per usual) or it can run on RStudio Server using a Docker Image + container. We have provided installation and run instructions following Nature's software checklist.

(3)

The authors should be consistent, using either 'serologic' or 'serological'.

Response: For consistency, we have changed all instances of "serological" in the manuscript to "serologic".